# Treatment heterogeneity of water, sanitation, hygiene, and nutrition interventions on child growth by environmental enteric dysfunction and pathogen status for young children in Bangladesh

**Zachary Butzin-Dozier**[1], **Yunwen Ji**[1], **Jeremy Coyle**[1], **Ivana Malenica**[1], **Elizabeth T. Rogawski McQuade**[2], **Jessica Anne Grembi**[3], **James A. Platts-Mills**[4], **Eric R. Houpt**[4], **Jay P. Graham**[1], **Shahjahan Ali**[5], **Md Ziaur Rahman**[6], **Mohammad Alauddin**[7], **Syeda L. Famida**[5], **Salma Akther**[5], **Md. Saheen Hossen**[5], **Palash Mutsuddi**[5], **Abul K. Shoab**[5], **Mahbubur Rahman**[5], **Md. Ohedul Islam**[5], **Rana Miah**[5], **Mami Taniuchi**[4], **Jie Liu**[8], **Sarah T. Alauddin**[7], **Christine P. Stewart**[9], **Stephen P. Luby**[10], **John M. Colford Jr.**[1], **Alan E. Hubbard**[1], **Andrew N. Mertens**[1☯], **Audrie Lin**[6☯]*

**1** School of Public Health, University of California, Berkeley, California, United States of America, **2** Rollins School of Public Health, Emory University, Atlanta, Georgia, United States of America, **3** Department of Veterinary and Biomedical Sciences, The Pennsylvania State University, University Park, Pennsylvania, United States of America, **4** School of Medicine, University of Virginia, Charlottesville, Virginia, United States of America, **5** International Centre for Diarrhoeal Disease Research, Dhaka, Bangladesh, **6** Department of Microbiology and Environmental Toxicology, University of California, Santa Cruz, California, United States of America, **7** Wagner College, Staten Island, New York, New York, United States of America, **8** School of Public Health, Qingdao University, Qingdao, China, **9** Institute for Global Nutrition, University of California, Davis, California, United States of America, **10** Division of Infectious Diseases and Geographic Medicine, Stanford University, Stanford, California, United States of America

☯ These authors contributed equally to this work.
* audrielin@ucsc.edu

## Abstract

### Background

Water, sanitation, hygiene (WSH), nutrition (N), and combined (N+WSH) interventions are often implemented by global health organizations, but WSH interventions may insufficiently reduce pathogen exposure, and nutrition interventions may be modified by environmental enteric dysfunction (EED), a condition of increased intestinal permeability and inflammation. This study investigated the heterogeneity of these treatments' effects based on individual pathogen and EED biomarker status with respect to child linear growth.

### Methods

We applied cross-validated targeted maximum likelihood estimation and super learner ensemble machine learning to assess the conditional treatment effects in subgroups defined by biomarker and pathogen status. We analyzed treatment (N+WSH, WSH, N, or control) randomly assigned in-utero, child pathogen and EED data at 14 months of age, and child HAZ at 28 months of age. We estimated the difference in mean child height for

**Data availability statement:** All study data, analysis scripts, and pre-registered analysis plan are available via Open Science Framework (https://osf.io/cg8dv/).

**Funding:** This work was supported by Global Development grants [OPPGD759, OPP1165144, and OPP1161946] from the Bill & Melinda Gates Foundation to the University of California, Berkeley and by the National Institute of Allergy and Infectious Diseases of the National Institutes of Health [grant numbers K01AI136885 to AL and K01AI182501 to ZB]. ZB, JAG, SA, MZR, MA, SLF, SA, MSH, PM, AKS, MR, MOI, RM, SA, CPS, SPL, MJC, AEH, ANM, and AL received funding for part of their salary and travel expenses from the research grants provided for this project by the Bill and Melinda Gates Foundation and AL received salary support from grant K01AI136885. icddr,b is grateful to the Governments of Bangladesh and Canada for providing core/unrestricted support. The funders had no role in study design, data collection and analysis, decision to publish, or preparation of the manuscript.

**Competing interests:** The authors have declared that no competing interests exist.

age Z-score (HAZ) under the treatment rule and the difference in stratified treatment effect (treatment effect difference) comparing children with high versus low pathogen/biomarker status while controlling for baseline covariates.

## Results

We analyzed data from 1,522 children who had a median HAZ of –1.56. We found that fecal myeloperoxidase (N+WSH treatment effect difference 0.0007 HAZ, WSH treatment effect difference 0.1032 HAZ, N treatment effect difference 0.0037 HAZ) and *Campylobacter* infection (N+WSH treatment effect difference 0.0011 HAZ, WSH difference 0.0119 HAZ, N difference 0.0255 HAZ) were associated with greater effect of all interventions on anthropometry. In other words, children with high myeloperoxidase or *Campylobacter* infection experienced a greater impact of the interventions on anthropometry. We found that a treatment rule that assigned the N+WSH (HAZ difference 0.23, 95% CI (0.05, 0.41)) and WSH (HAZ difference 0.17, 95% CI (0.04, 0.30)) interventions based on EED biomarkers and pathogens increased predicted child growth compared to the randomly allocated intervention.

## Conclusions

These findings indicate that EED biomarkers and pathogen status, particularly *Campylobacter* and myeloperoxidase (a measure of gut inflammation), may be related to the impact of N+WSH, WSH, and N interventions on child linear growth.

## Author summary

Water, sanitation, hygiene, and nutrition interventions are often implemented with the goal of improving child growth and development, but we lack information on what determines which children can benefit most from these interventions. Frequent infection, gut inflammation, and systemic inflammation may limit the effectiveness of nutrition interventions by preventing children from using nutrients effectively. On the other hand, water, sanitation, and hygiene interventions may prevent infections and thereby reduce inflammation. We sought to evaluate the impact of water, sanitation, hygiene, and nutrition interventions on child growth if these interventions were assigned based on biological markers of infection and inflammation. We found that children with *Campylobacter* infection and high myeloperoxidase (a measure of gut inflammation) experienced the greatest effect of the interventions on growth. These findings support the role of infection and inflammation in determining the effectiveness of water, sanitation, hygiene, and nutrition interventions in improving child growth.

## Introduction

Approximately 148 million children globally experience linear growth faltering, which may be a consequence of early-life undernutrition [1]. Research consistently shows a positive link between child growth and development, leading to the use of linear growth as a proxy for overall development [2,3]. In adulthood, children who experienced early-life growth faltering are more likely to experience low educational attainment and low income, although

interventions that improve child growth do not necessarily improve child development (and vice versa) [2–6]. Children of mothers who are stunted have an increased risk of experiencing stunting themselves, which can perpetuate the cycle of poverty [7].

## Water, sanitation, hygiene, and nutrition

Experts in public health and international development have identified water, sanitation, hygiene (WSH), nutrition (N), and combined (N+WSH) programs as potentially effective methods to improve child growth. WSH interventions aim to reduce children's exposure to pathogens, which can improve nutrient utilization by reducing malabsorption, inflammation, redirection of nutrients for immune response, and other symptoms associated with infection. [8] Nutrition interventions aim to directly provide nutrient supplementation.

Several observational studies indicated a positive relationship between household WSH interventions and child growth [9]. In contrast to these observational findings, three recent randomized controlled trials in rural populations from Kenya and Bangladesh (the WASH Benefits study) and Zimbabwe (SHINE trial) found that household WSH interventions did not improve child linear growth in a randomized context [7,10–12]. The lack of impact of these interventions may reflect an inability of these household interventions to sufficiently reduce pathogen exposure and environmental enteric dysfunction [10,13,14].

The WASH Benefits study found that nutritional supplementation led to modest improvements in child linear growth compared to control, which was consistent with previous studies [10,15,16]. The combined N+WSH intervention did not provide any additional benefit to child linear growth compared to the nutrition intervention alone [10]. This small and variable impact of nutrition interventions may be due to contextual underlying factors influencing participants' ability to respond to and benefit from nutrition interventions [10,17].

## Effect measure modification by EED and pathogens

The WASH Benefits study did not detect significant effect modification of interventions by child age, child sex, maternal education, maternal age, child parity, economic factors, or child hunger [10]. Although, it should be noted that lack of observed effect measure modification could be due to limited power or the study context [10]. Despite this lack of evidence of interaction, enteropathogen and environmental enteric dysfunction (EED) biomarker data may provide additional information on which subgroups of children, defined by pathogen infection/carriage status or biomarker levels, are amenable or resistant to intervention (Fig 1).

EED is a condition characterized by increased gut permeability, gut barrier disruption, increased gut and systemic inflammation, and is hypothesized to be caused by chronic exposure to pathogens [19,20]. Although clear diagnostic criteria for EED have not been established, several studies have speculated that it could be a key intermediate between poverty and growth impairment for children in low and middle-income countries [19,20]. Observational data and animal models have indicated that *Campylobacter* infection may contribute to EED [21]. Among young children in Bangladesh, small intestine bacterial overgrowth was associated with both intestinal inflammation, a key component of EED, and child growth impairment [22,23].

Previous analyses of the WASH Benefits Bangladesh study evaluated the impact of interventions on child pathogens and EED biomarkers. The investigators found that the nutrition intervention was associated with reduced fecal neopterin at 3 and 14 months of age, and all interventions reduced urinary lactulose and mannitol at 3 and 14 months [24]. Urinary lactulose and mannitol measure gut permeability [25]. At 28 months, contrary to a-priori hypotheses, WSH and nutrition interventions were associated with increased fecal myeloperoxidase

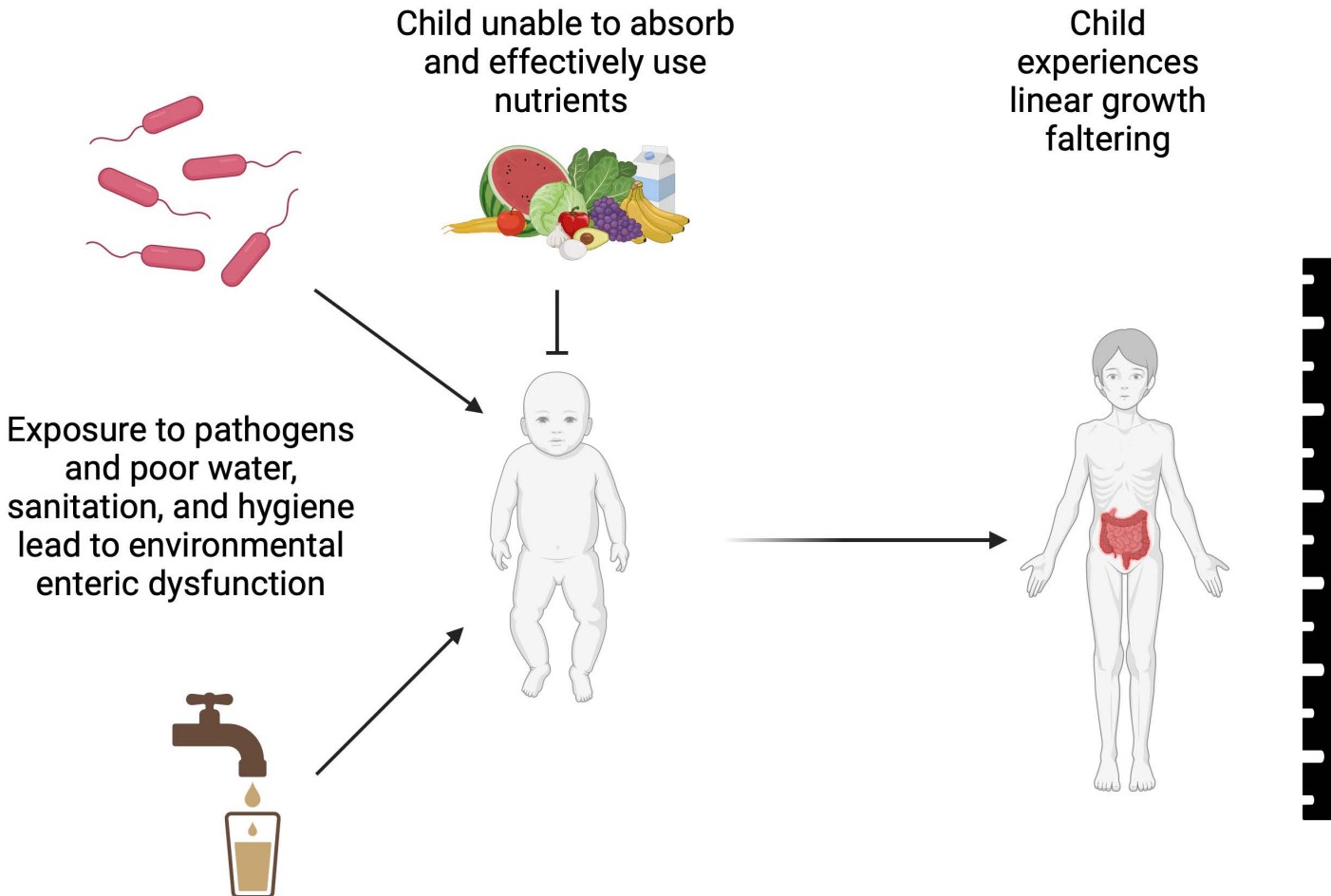

**Fig 1. Hypothesized relationships between water, sanitation, hygiene, nutrition, EED, and child growth.** Figure created with Biorender [18].

[24]. Although these findings at ages 3 and 14 months support N+WSH interventions' ability to reduce some EED biomarkers, the counterintuitive results at 28 months highlight uncertainty regarding the relationship between N+WSH interventions and presumed biomarkers for EED.

Investigators of the WASH Benefits study suggested that insufficient reduction of pathogen exposure could explain the null effects of WSH interventions on child linear growth [10]. Investigation of the relationships between N+WSH interventions and enteropathogens at Year 1 (age 14 months) in Bangladesh found that children who received WSH interventions had a lower prevalence and quantity of some individual viruses in their stools (norovirus, sapovirus, and adenovirus 40/41) compared to children in the control group, although investigators did not find a significant difference in bacteria, parasites, or stunting-related pathogens between these groups [13]. Furthermore, this study found that 99% of children at Year 1 had at least one enteropathogen [13]. At Year 2 (median age 28 months), investigators found that individual sanitation and hygiene interventions were associated with decreased *Giardia* infections compared to control and that drinking water but nutrition interventions were not associated with a decrease in *Giardia* infections compared to control [26]. Regarding soil-transmitted helminths, investigators found that the drinking water intervention was associated with reduced hookworm infection compared to control [27]. Lastly, analysis of interventions and fecal contamination found that drinking water and handwashing interventions reduced contamination of water and

food but did not reduce contamination of indirect pathways such as child hands and objects and that combined WSH interventions provided no additional benefit compared to individual interventions [28]. In the SHINE Trial, WSH interventions did not have an effect on bacterial, viral, or parasitic enteropathogen prevalence, although parasite concentrations were decreased in children's stools in the intervention arm [14]. These cumulative findings indicate that household WSH interventions can reduce child exposure to certain pathogens, although these results highlight heterogeneous relationships between interventions and individual pathogens.

Using data from the WASH Benefits Bangladesh study, analysis of treatment heterogeneity through estimating subgroup treatment effects and optimal treatment regimens can improve our understanding of child growth in low- and middle-income countries. Despite the widespread use of N+WSH interventions, investigators have found mixed evidence regarding these interventions' impact on child anthropometry [7,10,12]. This study will apply targeted machine learning methods to assess the conditional treatment effect of N+WSH, WSH, and N interventions on child linear anthropometry (child height for age Z score (HAZ)) by pathogen and EED biomarker status and explore rules for the optimal allocation of N+WSH, WSH, and N interventions in resource-constrained settings [29].

## Results

We analyzed data from 1,522 children with a median age of 28.1 months (IQR 26.8 months, 29.2 months), and our analytic sample had a median HAZ of −1.56 at Year 2 (Table 1).

### Relationships between pathogens or biomarkers and the conditional average treatment effect

To identify subgroups of children (based on EED biomarker and pathogen values) with the largest treatment effect, we analyzed the treatment effect comparing children with detection

**Table 1. Descriptive statistics of sample population.**

|  |  |  | *n* (%) or median (IQR) |
|---|---|---|---|
| Child |  | Female | 748 (49%) |
|  | Anthropometry (14 months, Year 1) | Length-for-age z-score | −1.41 (−2.06, −0.74) |
|  |  | Weight-for-age z-score | −1.31 (−2.01, −0.63) |
|  |  | Weight-for-length z-score | −0.89 (−1.55, −−0.21) |
|  |  | Head circumference-for-age z-score | −1.78 (−2.34, −1.12) |
|  | Anthropometry (28 months, Year 2) | Height-for-age z-score | −1.56 (−2.27, −0.94) |
|  |  | Weight-for-age z-score | −1.58 (−2.2, −0.93) |
|  |  | Weight-for-height z-score | −1.03 (−1.62, −0.38) |
|  |  | Head circumference-for-age z-score | −1.81 (−2.39, −1.2) |
|  | Diarrhea (14 months, Year 1) | Caregiver-reported 7-day recall | 192 (13%) |
|  | Diarrhea (28 months, Year 2) | Caregiver-reported 7-day recall | 114 (7%) |
| Mother |  | Age (years) | 23 (20, 27) |
|  | Anthropometry at enrollment | Height (cm) | 150.28 (146.81, 154.15) |
|  | Education | Schooling completed (years) | 7 (4, 9) |
|  | Depression at Year 1 | CESD-R score | 9 (6, 16) |
|  | Depression at Year 2 | CESD-R score | 10 (5, 17) |
|  | Perceived stress at Year 2 | Perceived Stress Scale score | 14 (10, 18) |
|  | Intimate partner violence | Any lifetime exposure | 835 (57%) |

IQR: Interquartile range; CESD-R: Center for Epidemiologic Studies Depression Revised scale.

(for pathogens, above median for EED biomarkers) versus non-detection (for pathogens, below median for EED biomarkers) values of each pathogen or biomarker.

We found that the following covariates were associated with a greater impact of N+WSH intervention on anthropometry under the optimal treatment rule: ETEC (correlation 0.45, treatment effect difference (comparing the treatment among children with ETEC detection to children without ETEC detection) 0.0019 HAZ), *Campylobacter jejuni/coli* (correlation 0.37, treatment effect difference 0.0016 HAZ), *Campylobacter* spp. (correlation 0.33, treatment effect difference 0.0011 HAZ), fecal REG1B (correlation 0.20, treatment effect difference 0.0005 HAZ), and fecal myeloperoxidase (correlation 0.15, treatment effect difference 0.0007 HAZ) (Table 2). The following covariates were associated with a lower impact of N+WSH intervention: aEPEC (correlation −0.41, treatment effect difference −0.0018 HAZ), EAEC (correlation −0.39, treatment effect difference −0.0015 HAZ), fecal alpha-1-antitrypsin (correlation −0.38, treatment effect difference −0.0013 HAZ), and EPEC (correlation −0.22, treatment effect difference −0.0009 HAZ).

The following EED biomarkers and pathogens were associated with greater WSH impact on HAZ under the optimal treatment rule: fecal myeloperoxidase (correlation 1.00, treatment effect difference 0.1032 HAZ), fecal alpha-1-antitrypsin (correlation 0.26, treatment effect difference 0.0259 HAZ), fecal REG1B (correlation 0.17, treatment effect difference 0.0105 HAZ), *Campylobacter jejuni/coli* (correlation 0.15, treatment effect difference 0.0143), *Campylobacter* spp. (correlation 0.13, treatment effect difference 0.0119 HAZ), EPEC (correlation 0.11, treatment effect difference 0.014 HAZ), and aEPEC (correlation 0.08, treatment effect difference 0.0099 HAZ) (Table 3). No EED biomarkers or pathogens were associated with lower WSH treatment effect.

The following EED biomarkers and pathogens were associated with greater impact of N on HAZ under the optimal treatment rule: *Campylobacter* spp. (correlation 0.17, treatment effect difference 0.0255 HAZ), *Campylobacter jejuni/coli* (correlation 0.15, treatment effect difference 0.0269 HAZ), fecal myeloperoxidase (correlation 0.06, treatment effect difference 0.0037 HAZ), and ETEC (correlation 0.05, treatment effect difference 0.0098 HAZ) (Table 4). EAEC

**Table 2. Biomarker and Pathogen Correlation with NWSH Conditional Average Treatment Effect.**

| Biomarker or pathogen | n | Correlation | Treatment effect (HAZ difference) at non-detection (pathogen) or below median (EED biomarker) | Treatment effect (HAZ difference) at detection (pathogen) or above median (EED biomarker) | Difference in Treatment effect (95% CI) |
|---|---|---|---|---|---|
| Any Enterotoxigenic *Escherichia coli* | 601 | 0.45 | −0.0006 | 0.0013 | 0.0019 (0.0018, 0.0021) |
| *Campylobacter jejuni/coli* | 602 | 0.37 | −0.0004 | 0.0013 | 0.0016 (0.0015, 0.0018) |
| *Campylobacter* spp. | 603 | 0.33 | −0.0004 | 0.0008 | 0.0011 (0.001, 0.0013) |
| REG 1B | 614 | 0.20 | −0.0003 | 0.0003 | 5e-04 (4e-04, 7e-04) |
| Myeloperoxidase | 615 | 0.15 | −0.0003 | 0.0004 | 7e-04 (5e-04, 8e-04) |
| Any enteropathogenic *Escherichia coli* | 602 | −0.22 | 0.0006 | −0.0003 | −-9e-04 (−0.001,−7e-04) |
| Alpha-1-antitrypsin | 615 | −0.38 | 0.0007 | −0.0006 | −0.0013 (−0.0014,−0.0011) |
| Enteroaggregative *Escherichia coli* | 603 | −0.39 | 0.0012 | −0.0002 | −0.0015 (−0.0016,−0.0013) |
| Atypical enteropathogenic *Escherichia coli* | 602 | −0.41 | 0.0006 | −0.0011 | −0.0018 (−0.0019,−0.0016) |

HAZ: Height for age Z-score; EED: Environmental enteric dysfunction; REG 1B: regenerating gene 1β.

Example interpretation (using Row 1: Any *Enterotoxigenic Escherichia coli* as an example): Among children with non-detection of any enterotoxigenic *Escherichia coli*, NWSH interventions were associated with a −0.0006 HAZ difference compared to control, while for children with detection of any enterotoxigenic *Escherichia coli*, NWSH interventions were associated with a 0.0013 HAZ difference compared to control (experiencing a 0.0019 HAZ greater treatment effect than children with non-detection of any enterotoxigenic *Escherichia coli*.

**Table 3. Biomarker and Pathogen Correlation with Conditional Average WSH Treatment Effect.**

| Biomarker or pathogen | n | Correlation | Treatment effect (HAZ difference) at non-detection (pathogen) or below median (EED biomarker) | Treatment effect (HAZ difference) at detection (pathogen) or above median (EED biomarker) | Difference in Treatment effect (95% CI) |
|---|---|---|---|---|---|
| Myeloperoxidase | 628 | 1.00 | −0.1973 | −0.0941 | 0.1032 (0.0988, 0.1075) |
| Alpha-1-antitrypsin | 629 | 0.26 | −0.1586 | −0.1328 | 0.0259 (0.0215, 0.0302) |
| REG 1B | 629 | 0.17 | −0.151 | −0.1405 | 0.0105 (0.0062, 0.0149) |
| *Campylobacter jejuni/coli* | 616 | 0.15 | −0.1486 | −0.1343 | 0.0143 (0.0099, 0.0186) |
| *Campylobacter* spp. | 618 | 0.13 | −0.1494 | −0.1375 | 0.0119 (0.0076, 0.0163) |
| Any Enteropathogenic *Escherichia coli* | 618 | 0.11 | −0.1535 | −0.1395 | 0.014 (0.0097, 0.0183) |
| Atypical enteropathogenic *Escherichia coli* | 618 | 0.08 | −0.1483 | −0.1384 | 0.0099 (0.0056, 0.0142) |
| Enteroaggregative *Escherichia coli* | 618 | 0.04 | −0.1522 | −0.1434 | 0.0088 (0.0045, 0.0132) |
| Any Enterotoxigenic *Escherichia coli* | 616 | 0.03 | −0.1452 | −0.1445 | 7e-04 (−0.0036, 0.005) |

HAZ: Height for age Z-score; EED: Environmental enteric dysfunction; REG 1B: regenerating gene 1β.

Example interpretation (using Row 1: myeloperoxidase as an example): Among children with below median concentration of myeloperoxidase, WSH interventions were associated with a −0.1973 HAZ difference compared to control, while for children with above median concentration of myeloperoxidase, WSH interventions were associated with a −0.0941 HAZ difference compared to control (experiencing a 0.1032 HAZ greater treatment effect than children with below median concentration of myeloperoxidase).

**Table 4. Biomarker and Pathogen Correlation with Nutrition Conditional Average Treatment Effect.**

| Biomarker or pathogen | n | Correlation | Treatment effect (HAZ difference) at non-detection (pathogen) or below median (EED biomarker) | Treatment effect (HAZ difference) at detection (pathogen) or above median (EED biomarker) | Difference in Treatment effect (95% CI) |
|---|---|---|---|---|---|
| *Campylobacter* spp. | 591 | 0.17 | −0.008 | 0.0175 | 0.0255 (0.0195, 0.0314) |
| *Campylobacter jejuni/coli* | 590 | 0.15 | −0.0049 | 0.0221 | 0.0269 (0.021, 0.0329) |
| Myeloperoxidase | 601 | 0.06 | 0.0038 | 0.0075 | 0.0037 (−0.0023, 0.0096) |
| Any enterotoxigenic *Escherichia coli* | 589 | 0.05 | −0.0015 | 0.0083 | 0.0098 (0.0039, 0.0158) |
| REG 1B | 600 | 0.04 | −0.0046 | 0.016 | 0.0207 (0.0147, 0.0266) |
| Atypical enteropathogenic *Escherichia coli* | 591 | −0.01 | 0.001 | 0.0026 | 0.0017 (−0.0043, 0.0076) |
| Any enteropathogenic *Escherichia coli* | 591 | −0.01 | 0.0053 | −0.0008 | −0.006 (−0.0119, −1e-04) |
| Alpha-1-antitrypsin | 601 | −0.05 | 0.0102 | 0.0007 | −0.0095 (−0.0155, −0.0036) |
| Enteroaggregative *Escherichia coli* | 591 | −0.07 | 0.0157 | −0.0024 | −0.0181 (−0.024, −0.0122) |

HAZ: Height for age Z-score; EED: Environmental enteric dysfunction; REG 1B: regenerating gene 1β.

Example interpretation (using Row 1: *Campylobacter* spp. as an example): Among children with non-detection of *Campylobacter* spp., the Nutrition intervention was associated with a −0.008 HAZ difference compared to control, while for children with detection of *Campylobacter* spp., the Nutrition intervention was associated with a 0.0175 HAZ difference compared to control (experiencing a 0.0255 HAZ greater treatment effect than children with non-detection of *Campylobacter* spp.).

(correlation −0.07, treatment effect difference −0.0181 HAZ) was associated with a lower impact of N intervention.

## Treatment allocation and predicted child anthropometry

When comparing the combined N+WSH (mean HAZ −1.62) and control (mean HAZ −1.54) arms (*n* = 756), an optimal treatment allocation assigned 331 children to N+WSH and 425 children to control (Table 5). The optimal treatment rule predicted greater child HAZ than the observed randomized intervention (observed HAZ −1.58 vs. optimal HAZ −1.35; optimal vs. observed HAZ difference 0.23 HAZ, 95% CI (0.05, 0.41)).

**Table 5. Average child anthropometry given optimized vs randomized treatment.**

| Study arms | n | Observed HAZ in treatment arm | Observed HAZ in control arm | Optimal allocation ratio (treatment: control) | Overall observed child HAZ | Optimized child HAZ | Predicted HAZ difference |
|---|---|---|---|---|---|---|---|
| N+WSH vs. control | 756 | −1.62 | −1.54 | (331:425) | −1.58 | −1.35 (−1.53, −1.17) | 0.23 (0.05, 0.41) |
| WSH vs. control | 752 | −1.69 | −1.54 | (9:743) | −1.62 | −1.45 (−1.58, −1.32) | 0.17 (0.04, 0.3) |
| Nutrition vs. control | 726 | −1.53 | −1.54 | (317:409) | −1.53 | −1.47 (−1.62, −1.31) | 0.07 (−0.09, 0.22) |

HAZ: Height for age Z-score; N+WSH: Combined nutrition, water, sanitation, and hygiene intervention; WSH: Water, sanitation, and hygiene intervention.

This table describes child anthropometry in each treatment arm, as well as predicted values under the optimal treatment rule (optimal treatment allocation ratio, child HAZ under optimal treatment rule, and difference between observed growth and optimized growth). Example interpretation (using Row 1: N+WSH vs. control as an example): From a sample of 756 children, we observed an average child HAZ of −1.62 in the N+WSH arm and −1.54 in the control arm (−1.58 HAZ overall). The optimal treatment rule would assign 331 children to treatment and 425 to control. Under this optimal treatment rule, the average child anthropometry (including treatment and control) would be −1.35 HAZ, which is 0.23 HAZ higher than the observed child growth.

In the contrast of WSH (mean HAZ −1.69) and control (mean HAZ −1.54) arms (n = 752), the optimal treatment rule assigned 9 children to receive WSH interventions and 743 children to receive control. The optimal treatment rule had greater predicted child HAZ than the observed randomized, static intervention (observed HAZ −1.62 vs. optimal HAZ −1.45; optimal vs. observed HAZ difference 0.17 HAZ, 95% CI (0.04, 0.3)).

After comparing the nutrition (mean HAZ −1.53) and control (mean HAZ −1.54) arms (n = 726), the optimal treatment rule assigned 317 children to receive the intervention and 409 children to be in the control group. The optimal treatment rule did not have significantly greater child HAZ compared to the observed randomized intervention (observed HAZ −1.53 vs. optimal HAZ −1.47; optimal vs. observed HAZ difference 0.07 HAZ, 95% CI (−0.09, 0.22).

## Post-hoc analysis

*Campylobacter* spp. and fecal myeloperoxidase were associated with a greater treatment effect across all three interventions (S1–S6 Figs). To improve interpretability, we dichotomized *Campylobacter* based on detection (non-zero value) vs. non-detection (zero value), while we dichotomized myeloperoxidase as high concentration (above median value) vs. low concentration (below median value). We conducted an exploratory evaluation of the combined impact of *Campylobacter* infection detection and high concentration of fecal myeloperoxidase on the conditional treatment effect under the optimal treatment rule (Table 6). The difference in treatment effect, comparing those with both *Campylobacter* spp. infection and high fecal myeloperoxidase to those with no *Campylobacter* spp. detection and below median fecal myeloperoxidase was 0.039 HAZ for N+WSH, 0.106 HAZ for WSH, and 0.022 HAZ for N.

As child diarrhea may be associated with child EED biomarker or pathogen status, we conducted a sensitivity analysis, excluding children with diarrhea at Year 1. In this sensitivity analysis, we found similar results to our main analysis (S1 Table).

## Discussion

Across all three interventions, fecal myeloperoxidase, an EED biomarker of gut inflammation, and *Campylobacter* were associated with a greater treatment effect [30,31]. In other words, children with *Campylobacter* infection and higher myeloperoxidase experienced the greatest benefit from the interventions, although the magnitude of these differences in treatment effects was typically small and not clinically significant. There was a greater N+WSH and WSH treatment effect among those with both *Campylobacter* infection and high fecal myeloperoxidase than those with either factor alone. The correlation of both *Campylobacter* and

**Table 6.  Conditional average treatment effect given levels of both *Campylobacter* and myeloperoxidase at 14 months.**

| Treatment arm | Treatment effect (HAZ difference) given *Campylobacter* non-detection and below median myeloperoxidase | Treatment effect (HAZ difference) given *Campylobacter* detection and above median myeloperoxidase | Difference in treatment effect (HAZ difference) | Difference 95% CI |
|---|---|---|---|---|
| N+WSH | −0.0007 | 0.0011 | 0.0018 | 0.0017, 0.002 |
| WSH | −0.1981 | −0.0917 | 0.1064 | 0.1021, 0.1107 |
| Nutrition | −0.0029 | 0.0187 | 0.0216 | 0.0157, 0.0276 |

N+WSH: Combined nutrition, water, sanitation, and hygiene intervention; WSH: Water, sanitation, and hygiene intervention; HAZ: for age Z-score.

myeloperoxidase biomarkers with the treatment effect indicates that these factors, implicated as a cause (*Campylobacter*) and a marker (myeloperoxidase) of EED, support that EED may play a role in the interventions' impact on growth [32]. More specifically, these findings support the importance of the gastrointestinal system (as *Campylobacter* is a leading cause of gastrointestinal illness and myeloperoxidase is a measure of gastrointestinal inflammation) as a key factor in determining the impact of these interventions on child anthropometry.[33] These results are consistent with previous findings that young children with *Campylobacter* infection may face an increased risk of growth impairment and are, therefore, a high-need group for intervention. A multi-site birth cohort study (MAL-ED) found that *Campylobacter* infection was highly prevalent and was associated with decreased child anthropometry in the first two years of life [34,35]. *Campylobacter* infections are endemic in settings where poultry is raised near the household (which is common in low and middle-income countries), and even asymptomatic infection is negatively associated with child growth [35–37]. *Campylobacter* alters the gut microbiota composition, disrupts the intestinal barrier, and can elicit chronic intestinal inflammation [38–43]. Across eight study sites in low-resource settings, MAL-ED found that breastfeeding, lack of access to WSH, and targeted antibiotic treatment were associated with *Campylobacter* infection, which in turn was associated with higher myeloperoxidase, higher α-1-antitrypsin, and lower fecal neopterin, which are key biomarkers of EED [35].

We found that myeloperoxidase, an EED marker of gut inflammation, was associated with a greater impact of N+WSH, WSH, and N interventions on child anthropometry. That is to say, children with higher intestinal inflammation were most protected by the interventions. This likely indicates children whose household environments were the most contaminated and interventions reduced but did not eliminate environmental exposures, as gut inflammation remained high despite continued intervention delivery. Previous meta-analyses have found that inflammation and WSH conditions modify the effects of nutrient supplementation on micronutrient status and anemia [15,44]. Regarding WSH, our findings were consistent with MAL-ED's findings that EED and inflammation likely mediated the relationship between infection and growth faltering [32]. In addition, MAL-ED investigators found that myeloperoxidase was associated with pathogen infection, and more specifically, that *Campylobacter* and myeloperoxidase were positively associated across all eight study sites [45].

There was a large disparity between the individual biomarker treatment effect differences (very small) and the overall shift in anthropometry under the optimal treatment regime (moderate). The optimal treatment regime takes all of the factors into account, while the treatment effect difference only looks at biomarkers and pathogens one at a time. Next, the treatment effect difference dichotomizes all of the biomarkers and pathogens, while the optimal

treatment regime incorporates their continuous values in whatever way is most informative to the optimal treatment regime. The disparity between these values highlights how a flexible nonparametric approach such as Targeted Machine Learning can outperform parametric specification of subgroups. These findings highlight the potential for targeted learning methods to identify and explore treatment heterogeneity within a study and for optimal treatment regime analysis to estimate the effects of targeting treatments to children who would benefit the most when resource constraints prevent intervening on all children.

These findings support the hypothesis that pathogen exposure and EED biomarkers are associated with HAZ faltering. Within rural Bangladesh, these effects were small, but they provide support for a biological mechanism.

## Strengths

The rich data source of the WASH Benefits Bangladesh EED substudy is a major strength of this analysis. This data source included *in utero* randomized interventions that were continued for two years after birth and robust collection of enrollment covariates, EED biomarkers, pathogens, and anthropometry outcomes across multiple timepoints. Furthermore, the statistical methods applied here allow us to flexibly assess relationships between multiple covariates, exposures, and outcomes while making minimal parametric assumptions.

The analysis methods are a second major strength of this study. We used targeted maximum likelihood estimation, which is maximally efficient in finite samples and doubly-robust [46,47]. Assessment of optimal individualized treatment effects allows us to evaluate the relationships between pathogen exposure, EED, and intervention effects without making parametric assumptions [48–54]. Given the complex relationships between these biomarker and pathogen data, these targeted learning methods allow flexible modeling of complex relationships without requiring parametric assumptions regarding relationships between interventions, biomarkers, pathogens, and child anthropometry that would inevitably be violated.

## Limitations

One limitation of this study arises from using post-intervention biomarkers and pathogens, as no baseline EED biomarkers or pathogens were measured because infants were *in utero* at the time of randomization. Conditioning on these post-intervention nodes potentially introduces confounding and bias. We accounted for this possible confounding by adjusting for additional baseline covariate information related to family health and socioeconomic status and by excluding pathogens and EED biomarkers that were associated with the interventions in previous analyses of this sample (i.e., potential mediators or colliders), although residual confounding or bias may be present. Using growth velocity evaluated after biomarker assessment, rather than HAZ, as an outcome of interest may alleviate these concerns in some settings, but it would have considerable drawbacks in our study context. As our interventions were delivered at birth, we do not have pre-intervention anthropometric assessments, which would be required to assess the full impact of the interventions on growth velocity. Furthermore, as we randomized in-utero, we can infer that anthropometric characteristics were balanced at baseline. While it would be feasible to calculate growth velocity as the change in growth between Year 1 and Year 2, this presents additional inferential challenges, as children with early life growth faltering may experience subsequent "catchup growth" due to regression to the mean (convergence towards average trajectory) and/or due to post-birth biological recovery [55,56]. Therefore, given the effective randomization of this study, we selected HAZ as our outcome of interest.

In the future, we hope to analyze biological samples that were collected from these children at a younger age (4–8 months) to further evaluate these relationships. Furthermore, future

studies should evaluate these relationships by assessing biomarker and pathogen status before randomization. The external validity of these findings is also limited due to the trial context, in which participating households received extensive follow-up and monitoring and may not reflect the experience of these interventions in the target population's context.

The small or null overall effects of the study interventions are another limitation. In the presence of a null overall effect, to detect subpopulations with a significant effect, there must be equivalent populations with a deleterious effect or much larger populations with a null effect. In contrast, optimal treatment regime analysis in a population with a greater treatment effect will have much greater power to detect subpopulations of interest. While the trial reported that N and N+WSH interventions led to a modest improvement in HAZ [10], these effects were not seen for this subsample. This may be why more than half of the children were assigned to control rather than the interventions in the optimal treatment regime, which should be taken as a finite sample limitation of a trial with null effects on children within the small substudy. Follow up evaluation of these relationships in a separate population may provide insight on the replicability of these findings. The greater magnitude of the treatment effect difference given *Campylobacter* detection and above median fecal myeloperoxidase versus *Campylobacter* non-detection and below median myeloperoxidase for WSH, compared to N+WSH, may be attributable to finite sample bias, in which the observed HAZ in the WSH arm was lower than the other three treatment arms. This large treatment effect difference for WSH vs. control may be attributable to the wide variation in anthropometry observed in the WSH arm compared to other treatment arms, which makes this group amenable to an optimized intervention.

### Future directions

These findings support the application and evaluation of interventions that aim to reduce exposure to and infection by pathogens, such as *Campylobacter*, as well as interventions that seek to reduce intestinal inflammation. Evaluations of these interventions should evaluate their direct impacts on these biomarkers and pathogens as well as their indirect impacts on child anthropometry.

We found that the interventions had the greatest effect in children with a high burden of pathogens and EED biomarkers. Future evaluations that consistently identify biomarkers and pathogens associated with lower treatment effect (i.e., resistance to treatment) could indicate the need for co-interventions. For example, certain types of persistent bacterial infection (e.g., *Mycobacterium tuberculosis* or *Salmonella typhi*) may not be responsive to WSH interventions, and may require additional medical intervention [57–59]. In these cases, co-interventions, such as antibiotic treatment, may supplement interventions in order to ameliorate these conditions and improve N+WSH, WSH, or N intervention effectiveness [57].

We focused our interpretation on *Campylobacter* and myeloperoxidase, which demonstrated consistent correlations (in terms of direction) with the conditional average treatment effect (CATE) across interventions. Our analysis of individual biomarkers' and pathogens' correlations with the conditional treatment effect provided some evidence of effect heterogeneity being associated with factors beyond *Campylobacter* and fecal myeloperoxidase, although the lack of consistency of these observations across similar interventions (e.g., N+WSH versus WSH) led us to believe that these relationships may be spurious. On the other hand, it is plausible that these unique correlations across similar biomarkers and pathogens point to unique actions of related covariates or unique mechanisms of combined versus individual interventions, respectively. Future studies could incorporate cluster analysis methods to assess the combined role of related biomarkers and pathogens on treatment effectiveness.

## Conclusion

The cumulative results here indicate that EED and pathogens may be related to N+WSH, WSH, and N interventions' impact on child anthropometry. In particular, we found that *Campylobacter* infection and high myeloperoxidase were associated with a greater effect of N+WSH (treatment effect difference 0.039 HAZ), WSH (treatment effect difference 0.106 HAZ), and N (treatment effect difference 0.022 HAZ) interventions on child HAZ at 28 months. These findings are consistent with the observational results of the MAL-ED study [32,35,60]. This information regarding the relationships between pathogens, EED biomarkers, and treatment effectiveness highlights biological mechanisms that may indicate an individual's ability to respond to N+WSH, WSH, and N interventions. These results may help distinguish what defines a responsive versus nonresponsive individual to these interventions and should motivate future etiological research that seeks to estimate the causal impact of EED and pathogen burden on intervention effectiveness.

## Methods

### Ethics

The primary caregiver of each child provided written informed consent prior to enrollment. Human subjects protection committees at International Centre for Diarrhoeal Disease Research, Bangladesh (icddr,b), the University of California, Berkeley, and Stanford University approved the study protocols. The parent trial was registered at ClinicalTrials.gov (NCT01590095) and a safety monitoring committee convened by icddr,b oversaw the study.

### Study design, participants, and interventions

This analysis involves data from a substudy of the WASH Benefits Bangladesh randomized controlled trial. The trial randomized pregnant mothers and their children to receive one of six interventions – water treatment, sanitation, handwashing, nutrition (N), combined water treatment, sanitation and handwashing (WSH), and combined nutrition plus WSH (N+WSH), or control (see enrollment flowchart in S7 Fig and study timeline in S8 Fig) [10]. In a substudy focused on the evaluation of EED, investigators assessed additional biomarker data in a subset of children in four of the study arms – N, WSH, N+WSH, and control (with an allocation ratio of 1:1:1:1) [24]. The behavioral components of these interventions included treating drinking water for index households, which included children less than 3 years of age (water), using latrines and child potty in addition to removing animal feces from the compound (sanitation), washing hands with soap before preparing food and after defecating or contacting child feces (hygiene), and practicing age-appropriate nutrition practices from pregnancy up until two years of age and using small-quantity lipid-based nutrient supplements for children six months to two years of age (nutrition) [61]. Promoters were instructed to visit study compounds at least once per week for the first six months and then once every two weeks for the following 1.5 years (until child age 24 months). The intervention hardware and consumables were provided free of charge and replenished by promoters as needed throughout the study period (additional details on interventions can be found in S1 Text).

Investigators followed the cohort of children for approximately 2.5 years after birth. Due to logistical challenges regarding specimen collection and transportation, it was not feasible to retain the geographic matching of the parent trial in this subset. The trial was conducted in contiguous rural subdistricts in Gazipur, Mymensingh, Tangail, and Kishoreganj districts of Bangladesh. The trial enrolled women in their first or second trimester of pregnancy (additional information on recruitment and eligibility can be found in S2 Text) [61].

**Covariates.**  Although randomization of participants led to a balanced distribution of covariates between study arms, this analysis conditioned on post-randomization biomarker values, leading to the possibility of collider stratification bias. In addition to the biomarkers and pathogens included in the treatment rule, our analyses adjusted for child sex, birth order, number of children under 18 years of age in the household, number of individuals in the compound (group of nearby houses), household wall material, household wealth (first principal component of a principal components analysis incorporating household assets), maternal age and height, age in days at urine and stool assessments, month of urine and stool assessments, and age at anthropometry assessment. We considered and tested potential confounders using super learner and cross-validated targeted maximum likelihood estimation. The full list of baseline and time-varying covariates can be found in S3 Text.

**Biomarkers and pathogens.**  *EED biomarkers:* The EED measures included in this study were fecal alpha-1-antitrypsin, myeloperoxidase, and REG1B, which we measured at median ages 3 months and 14 months. These measures are markers of intestinal permeability (alpha-1-antitrypsin), inflammation (myeloperoxidase), and intestinal repair (REG1B) [30]. We excluded EED biomarkers (fecal neopterin and urinary lactulose and mannitol) that were associated with the interventions of interest in a previous analysis of this sample and therefore were potential mediators of the exposure-outcome relationship [24].

To reduce inter-laboratory variation, all fecal samples were assayed by the same research team member at the International Centre for Diarrhoeal Disease Research, Bangladesh (icd-dr,b) laboratory. Laboratory methods are included in S4 Text and were published previously [13,24].

*Pathogens:* We included six pathogens in our final analysis: *Campylobacter jejuni/coli*, enteroaggregative *Escherichia coli* (EAEC), any enterotoxigenic *E. coli* (ETEC), atypical enteropathogenic *E. coli* (aEPEC), any enteropathogenic *E. coli* (EPEC, including both typical or atypical), and *Campylobacter* spp. (which includes *C. jejuni, C. coli,* and other *Campylobacter* species). Relative concentrations of pathogens were assessed at 14 months in feces using quantitative polymerase chain reaction (qPCR) via TaqMan array card [13,62,63]. We excluded three pathogens (norovirus, sapovirus, and adenovirus 40/41) that were reduced by the interventions of interest in a previous analysis of this sample and therefore were potential mediators of the exposure-outcome relationship [13]. We excluded an additional 25 pathogens due to high missingness or near-zero variance. We quantified pathogens using quantification cycle, where one unit corresponded to twice the pathogen quantity. The analytical limit of detection was quantification cycle 35, values above this were considered non-detects [64]. We standardized these measures using the efficiency of per-sample extraction/amplification. The full list of pathogens is included in S2 Table.

A single infection event is unlikely to elicit HAZ impairment in itself, but repeated exposure to pathogens and chronic disruptions such as EED are associated with delayed growth [32,35,65]. This analysis assumes that the detection of pathogens and EED biomarkers at 14 months is a proxy for chronic exposure to these factors throughout early childhood.

**Outcomes.**  The anthropometry outcome was height for age Z-score (HAZ) assessed at Year 2 (median age 28 months). Following standard protocols for anthropometric outcomes measurement [66,67], pairs of trained anthropometrists measured child anthropometry (accurate to 0.1 cm) in triplicate to calculate median HAZ using 2006 WHO child growth standards [10]. We measured recumbent length when child was age < 24 months, and we measured standing height when child was age > 24 months.

**Analyses.**  These analyses assessed the conditional average treatment effect (CATE) and mean under the optimal individualized treatment regime using a targeted learning approach [48]. A static treatment approach as used in the WASH Benefits primary analysis, in which

treatment is randomly assigned, aims to assess the average effect of the interventions in the entire study population (i.e., interventions are not targeted based on individual covariate information) [10,48]. In contrast, an optimal treatment regime analysis assesses the impact of the intervention given (or, conditional on) individual covariate status [10,48]. In these analyses, the individual covariate information was child pathogen and EED biomarker status.

We used cross-validated targeted maximum likelihood estimation, which we fit using Super Learner ensemble machine learning in order to estimate the optimal individualized treatment regime and the outcome as the mean under the optimal individualized treatment [68]. First, we estimated the outcome regression function and propensity score (treatment mechanism) using Super Learner. Super Learner is an ensemble machine learning approach that creates a convex combination of candidate algorithms in order to maximize model fit [69,70]. Super Learner is grounded in statistical optimality theory, and guarantees that it will perform at least as well as the best candidate algorithm with a sufficiently large sample size. In our learner list for the treatment mechanism, we included the least absolute shrinkage and selection operator (LASSO) penalized regressions, random forests, the simple mean, and generalized linear models, and we used non-negative least squares to construct the final ensemble (the meta-learner) [48].

Next, we used the doubly-robust augmented inverse probability weighting to transform the outcome to a random variable that has the CATE (i.e., the treatment effect specific to each individual's set of covariates) as its mean and regressed this transformed outcome to assess treatment heterogeneity using targeted maximum likelihood estimation via the R package "tmle3mopttx" [71]. Targeted maximum likelihood estimation reduces bias and yields an interpretable measure of association (in this case, the average treatment effect) [48,72–75]. Specifically, we estimated the function of the individualized outcome by regressing this contrast on biomarker status using Super Learner with a non-negative least squares loss function based on the Lawson-Hanson algorithm. As these analyses assess the impact of the randomized intervention (the treatment mechanism), the doubly-robust nature of this estimator will ensure asymptotically consistent estimation of the CATE even if the outcome regression is not consistently estimated [48].

Finally, we use the estimate of the CATE function to derive an optimal individualized treatment rule where we would treat a maximum of 50% of individuals with the greatest CATE. Though providing optimal treatment to all children is desirable, in a resource constrained setting, one might also be interested to limit the intervention to the children most likely to benefit from the intervention (i.e., have the greatest CATE). In order to assess the impact of the individualized treatment regime in resource-constrained settings (i.e., preventing all children from being allocated to intervention), we restricted the maximum allocation to treatment in each binary (treatment to control) contrast to be no more than 50%, which is approximately equivalent to the original trial's allocation ratio (1:1:1:1). If less than 50% of individuals in a single binary (treatment to control) contrast have a positive CATE (beneficial effect of treatment), then the optimal treatment rule will assign all individuals with a positive CATE to intervention. If more than 50% of individuals in a single contrast have a positive CATE, the optimal treatment rule will only assign the 50% of individuals with the greatest CATE to intervention.

To assess the role of each biomarker or pathogen in the optimal treatment rule, we evaluated Pearson's correlation between each of these covariates and the CATE. To contextualize the magnitude of these relationships, we estimated the difference in subgroup treatment effect between children with detection (for pathogens, above median for EED biomarkers) versus non-detection (for pathogens, below median for EED biomarkers) pathogen and EED biomarker status. While the optimal treatment rule flexibly incorporated continuous

values of these biomarkers, we used binary transformations of these values to assess variable importance in order to improve interpretability. Furthermore, while the optimal treatment rule assessed the combined role of these biomarkers, our assessments of variable importance assessed each biomarker individually to improve interpretability. The difference between these two subgroup effects is hereafter referred to as "treatment effect difference."

*Covariate screening:* We screened all covariates for missingness, excluding all covariates with missingness greater than 30% and median-imputing all other missing covariate data. We only included observations for which the primary outcome, HAZ at 28 months, was observed. We also excluded variables with near zero variance, which we defined as covariates with a frequency ratio (ratio of most frequent value to second most frequent value) greater than 2 and a percent of unique values less than 20%, using the R package "caret" (version 6.0–92) [76]. The analysis plan was publicly pre-registered on Open Science Framework, and all data and analysis scripts are publicly available (https://osf.io/cg8dv/). EED markers assessed at 3 months were excluded due to high missingness (>30%). The full list of excluded covariates and reasons for exclusion are defined in S2 Table.

## Supporting information

**S1 Fig. Correlation of individual N+WSH conditional treatment effect and myeloperoxidase concentration.** N+WSH, combined nutrition, water, sanitation, and hygiene intervention.
(DOCX)

**S2 Fig. Correlation of individual N+WSH conditional treatment effect and Campylobacter concentration.** N+WSH, combined nutrition, water, sanitation, and hygiene intervention.
(DOCX)

**S3 Fig. Correlation of individual WSH conditional treatment effect and myeloperoxidase concentration.** WSH, water, sanitation, and hygiene intervention.
(DOCX)

**S4 Fig. Correlation of individual WSH conditional treatment effect and *Campylobacter* concentration.** WSH, water, sanitation, and hygiene intervention.
(DOCX)

**S5 Fig. Correlation of individual WSH conditional treatment effect and myeloperoxidase concentration.** WSH, water, sanitation, and hygiene intervention.
(DOCX)

**S6 Fig. Correlation of individual nutrition conditional treatment effect and *Campylobacter* concentration.**
(DOCX)

**S1 Table. Biomarker and Pathogen Correlation with NWSH Conditional Average Treatment Effect, Excluding Children with Diarrhea at Year 1.**
(DOCX)

**S7 Fig. Enrollment flowchart.**
(DOCX)

**S8 Fig. Study timeline.** Figure created using BioRender [1].
(DOCX)

**S1 Text. Study interventions.**
(DOCX)

**S2 Text. Inclusion criteria.**
(DOCX)

**S3 Text. Adjustment covariates.**
(DOCX)

**S4 Text. Laboratory methods.**
(DOCX)

**S2 Table. Pathogens, EED biomarkers, covariates, and reasons for exclusion** [6]**.**
(DOCX)

**S1 Checklist. CONSORT Checklist.**
(DOCX)

## Acknowledgments

We thank the families who participated in the WASH Benefits study and the incredible icddr,b staff for their valuable contributions.

The content is solely the responsibility of the authors and does not necessarily represent the official views of the National Institutes of Health.

## Author contributions

**Conceptualization:** Zachary Butzin-Dozier, Elizabeth T. Rogawski McQuade, Jessica Anne Grembi, James A. Platts-Mills, Eric R. Houpt, Mahbubur Rahman, Christine P. Stewart, John M. Colford, Jr., Alan E. Hubbard, Audrie Lin.

**Data curation:** Audrie Lin.

**Formal analysis:** Zachary Butzin-Dozier, Yunwen Ji, Jeremy Coyle, Ivana Malenica, Alan E. Hubbard, Andrew N. Mertens.

**Funding acquisition:** Stephen P. Luby, John M. Colford, Jr..

**Investigation:** Zachary Butzin-Dozier, Elizabeth T. Rogawski McQuade, Jessica Anne Grembi, Christine P. Stewart, Stephen P. Luby, Audrie Lin.

**Methodology:** Zachary Butzin-Dozier, Jeremy Coyle, Jessica Anne Grembi, John M. Colford, Jr., Alan E. Hubbard, Andrew N. Mertens.

**Project administration:** John M. Colford, Jr., Alan E. Hubbard, Audrie Lin.

**Resources:** Stephen P. Luby, John M. Colford, Jr., Alan E. Hubbard, Audrie Lin.

**Software:** Jeremy Coyle, Ivana Malenica, Alan E. Hubbard.

**Supervision:** Christine P. Stewart, John M. Colford, Jr., Alan E. Hubbard, Audrie Lin.

**Visualization:** Zachary Butzin-Dozier, Andrew N. Mertens.

**Writing – original draft:** Zachary Butzin-Dozier, Yunwen Ji.

**Writing – review & editing:** Zachary Butzin-Dozier, Yunwen Ji, Jeremy Coyle, Ivana Malenica, Elizabeth T. Rogawski McQuade, Jessica Anne Grembi, James A. Platts-Mills, Eric R. Houpt, Jay P. Graham, Shahjahan Ali, Md Ziaur Rahman, Mohammad Alauddin, Syeda L. Famida, Salma Akther, Md. Saheen Hossen, Palash Mutsuddi, Abul K. Shoab, Mahbubur Rahman, Md. Ohedul Islam, Rana Miah, Mami Taniuchi, Jie Liu, Sarah T. Alauddin, Christine P. Stewart, Stephen P. Luby, John M. Colford, Jr., Alan E. Hubbard, Andrew N. Mertens, Audrie Lin.

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
