## [Decision Letter · Decision Letter 0]

30 Sep 2024

Dear Dr. Lin,

Thank you very much for submitting your manuscript "Treatment Heterogeneity of Water, Sanitation, Hygiene, and Nutrition Interventions on Child Growth by Environmental Enteric Dysfunction and Pathogen Status for Young Children in Bangladesh" for consideration at PLOS Neglected Tropical Diseases. As with all papers reviewed by the journal, your manuscript was reviewed by members of the editorial board and by several independent reviewers. In light of the reviews (below this email), we would like to invite the resubmission of a significantly-revised version that takes into account the reviewers' comments. 

The authors have contributed results of an informative study detailing the potential benefits of WASH interventions on growth and enteropathogens in children from four endemic districts of Bangladesh. There are several areas which could benefit from a more clear and concise description regarding how the methods are linked to results, particularly regarding the biomarkers and enteropathogen correlates. 

We cannot make any decision about publication until we have seen the revised manuscript and your response to the reviewers' comments. Your revised manuscript is also likely to be sent to reviewers for further evaluation.

Sincerely,

Kirkby D Tickell, MBBS BSc

Academic Editor

Amy Gilbert

Section Editor

Reviewer's Responses to Questions

**Key Review Criteria Required for Acceptance?**

**Methods**

-Are the objectives of the study clearly articulated with a clear testable hypothesis stated?

-Is the study design appropriate to address the stated objectives?

-Is the population clearly described and appropriate for the hypothesis being tested?

-Is the sample size sufficient to ensure adequate power to address the hypothesis being tested?

-Were correct statistical analysis used to support conclusions?

-Are there concerns about ethical or regulatory requirements being met?

Reviewer #1: (No Response)

Reviewer #2: The authors methods more commonly used in a different field of medicine to ask an interesting question about WASH and nutrition interventions. The authors clearly explain technical methods. The study population and sample size were appropriate for the research question. I am not familiar with the statistical methods used and advise a technical review by someone who is.

**Results**

-Does the analysis presented match the analysis plan?

-Are the results clearly and completely presented?

-Are the figures (Tables, Images) of sufficient quality for clarity?

Reviewer #1: (No Response)

Reviewer #2: The results are clearly presented and match the analysis.

**Conclusions**

-Are the conclusions supported by the data presented?

-Are the limitations of analysis clearly described?

-Do the authors discuss how these data can be helpful to advance our understanding of the topic under study?

-Is public health relevance addressed?

Reviewer #1: (No Response)

Reviewer #2: The authors are careful with the conclusions made from this analysis given the limitations. I do believe the data presented supports the conclusion as additional support to the hypothesis that EED and pathogen status may modify the effect of WASH interventions.

**Editorial and Data Presentation Modifications?**

Reviewer #1: (No Response)

Reviewer #2: Minor: in the Post-hoc Analysis (pg 11) impact of campylobactor infection is described as any detection vs no detection, I suggest changing the language to be consistent with other analyses as high burden or high campy detection-if that is correct. Or use pathogen detection and no detection throughout. 

Minor: If there is space for a figure, it might be helpful to show a timeline of follow up from birth (intervention randomization) to 2 (growth measure) with timepoint of pathogen and biomarker testing. I found my self looking for this information more than once to familiarize myself with the timeline. 

Minor: The authors describe why the campylobactor and myeloperoxidase results are plausible. I am curious if you have any idea why campylobacter and myeloperoxidase stood out and other pathogens did not. Were you surprised by this?

**Summary and General Comments**

Reviewer #1: (No Response)

Reviewer #2: The authors share an interesting application of methods used in precision medicine to the aid the further understanding of WASH benefits study results. The manuscript is a clear presentation of technical methods not commonly used in international health/WASH/Growth faltering research.

PLOS authors have the option to publish the peer review history of their article (what does this mean? ). If published, this will include your full peer review and any attached files.

**Do you want your identity to be public for this peer review?** For information about this choice, including consent withdrawal, please see our Privacy Policy .

Reviewer #1: No

Reviewer #2: No
---

## [Decision Letter · Decision Letter 1]

25 Dec 2024

PNTD-D-24-00822R1Treatment Heterogeneity of Water, Sanitation, Hygiene, and Nutrition Interventions on Child Growth by Environmental Enteric Dysfunction and Pathogen Status for Young Children in BangladeshPLOS Neglected Tropical Diseases Dear Dr. Lin, Thank you for submitting your manuscript to PLOS Neglected Tropical Diseases. After careful consideration, we feel that some minor revisions are still required to the manuscript to fully meet PLOS Neglected Tropical Diseases's publication criteria as it currently stands. Therefore, we invite you to submit a revised version of the manuscript that addresses the points raised during the re-review. In addition to the comments attached to this decision, the reviewer also noted that if a ct cut-off other than <35 was used, they believe it would be appropriate to conduct a sensitivity analysis using that cutoff. Please also consider this comment in your revisions.  Please submit your revised manuscript within 30 days Jan 24 2025 11:59PM. If you will need more time than this to complete your revisions, please reply to this message or contact the journal office at plosntds@plos.org. Please include the following items when submitting your revised manuscript:* A rebuttal letter that responds to each point raised by the editor and reviewer(s). You should upload this letter as a separate file labeled 'Response to Reviewers '. This file does not need to include responses to any formatting updates and technical items listed in the 'Journal Requirements' section below.  * To expedite the review of your response, please copy and paste the revised test into your letter, rather than reference to revised manuscript.  * A marked-up copy of your manuscript that highlights changes made to the original version. You should upload this as a separate file labeled 'Revised Manuscript with Track Changes '. * An unmarked version of your revised paper without tracked changes. You should upload this as a separate file labeled 'Manuscript '. If you would like to make changes to your financial disclosure, competing interests statement, or data availability statement, please make these updates within the submission form at the time of resubmission. Guidelines for resubmitting your figure files are available below the reviewer comments at the end of this letter. We look forward to receiving your revised manuscript. Kind regards,Kirkby D Tickell, MBBS BScAcademic EditorPLOS Neglected Tropical Diseases Kirkby TickellAcademic EditorPLOS Neglected Tropical Diseases

Shaden Kamhawi

co-Editor-in-Chief

Paul Brindley

co-Editor-in-Chief

**Journal Requirements:**

1) Thank you for including an Ethics Statement for your study. Please include:

i) A statement that formal consent was obtained (must state whether verbal/written) and for child participants, the statement must declare that formal consent was obtained from the parent/guardian.].

2) Please include in the legends of Figure 1 and Supplemental Material 9 that the figures were created with BioRender.com. Please note that the licenses should not be uploaded with the item type "Supplementary Information."

3)The following file is currently uploaded as file type 'Other', which is not viewable by the reviewers: CONSORT Checklist.docx. Please change the file type to 'Supporting Information' and include a legend in the manuscript if you wish it to be included in review.

**Comments to the Authors:**  Please note the the review is uploaded as an attachment. **Reviewers' comments:**Reviewer's Responses to Questions

**Key Review Criteria Required for Acceptance?**

**Methods**

-Are the objectives of the study clearly articulated with a clear testable hypothesis stated?

-Is the study design appropriate to address the stated objectives?

-Is the population clearly described and appropriate for the hypothesis being tested?

-Is the sample size sufficient to ensure adequate power to address the hypothesis being tested?

-Were correct statistical analysis used to support conclusions?

-Are there concerns about ethical or regulatory requirements being met?

Reviewer #1: (No Response)

**Results**

-Does the analysis presented match the analysis plan?

-Are the results clearly and completely presented?

-Are the figures (Tables, Images) of sufficient quality for clarity?

Reviewer #1: (No Response)

**Conclusions**

-Are the conclusions supported by the data presented?

-Are the limitations of analysis clearly described?

-Do the authors discuss how these data can be helpful to advance our understanding of the topic under study?

-Is public health relevance addressed?

Reviewer #1: (No Response)

**Editorial and Data Presentation Modifications?**

Reviewer #1: (No Response)

**Summary and General Comments**

Reviewer #1: (No Response)

PLOS authors have the option to publish the peer review history of their article (what does this mean? ). If published, this will include your full peer review and any attached files.

**Do you want your identity to be public for this peer review?** For information about this choice, including consent withdrawal, please see our Privacy Policy .

Reviewer #1: No

[NOTE: If reviewer comments were submitted as an attachment file, they will be attached to this email and accessible via the submission site. Please log into your account, locate the manuscript record, and check for the action link "View Attachments". If this link does not appear, there are no attachment files. **Figure resubmission:** While revising your submission, please upload your figure files to the Preflight Analysis and Conversion Engine (PACE) digital diagnostic tool, https://pacev2.apexcovantage.com/. PACE helps ensure that figures meet PLOS requirements. To use PACE, you must first register as a user. Registration is free. Then, login and navigate to the UPLOAD tab, where you will find detailed instructions on how to use the tool. If you encounter any issues or have any questions when using PACE, please email PLOS at figures@plos.org. Please note that Supporting Information files do not need this step. If there are other versions of figure files still present in your submission file inventory at resubmission, please replace them with the PACE-processed versions. **Reproducibility:** To enhance the reproducibility of your results, we recommend that authors of applicable studies deposit laboratory protocols in protocols.io, where a protocol can be assigned its own identifier (DOI) such that it can be cited independently in the future. Additionally, PLOS ONE offers an option to publish peer-reviewed clinical study protocols. Read more information on sharing protocols at https://plos.org/protocols?utm_medium=editorial-email&utm_source=authorletters&utm_campaign=protocols

---

## [Editor Report · Decision Letter 2]

30 Jan 2025

Dear Dr. Lin,

We are pleased to inform you that your manuscript 'Treatment Heterogeneity of Water, Sanitation, Hygiene, and Nutrition Interventions on Child Growth by Environmental Enteric Dysfunction and Pathogen Status for Young Children in Bangladesh' has been provisionally accepted for publication in PLOS Neglected Tropical Diseases.

Best regards,

Kirkby D Tickell, MBBS BSc

Academic Editor

Kirkby Tickell

Academic Editor

Shaden Kamhawi

co-Editor-in-Chief

Paul Brindley

co-Editor-in-Chief

---

## [Editor Report · Acceptance letter]

Dear Dr. Lin,

We are delighted to inform you that your manuscript, "Treatment Heterogeneity of Water, Sanitation, Hygiene, and Nutrition Interventions on Child Growth by Environmental Enteric Dysfunction and Pathogen Status for Young Children in Bangladesh," has been formally accepted for publication in PLOS Neglected Tropical Diseases.

Best regards,

Shaden Kamhawi

co-Editor-in-Chief

Paul Brindley

co-Editor-in-Chief
